# Effects of Dietary Linoleic Acid on Blood Lipid Profiles: A Systematic Review and Meta-Analysis of 40 Randomized Controlled Trials

**DOI:** 10.3390/foods12112129

**Published:** 2023-05-25

**Authors:** Qiong Wang, Hui Zhang, Qingzhe Jin, Xingguo Wang

**Affiliations:** State Key Lab of Food Science and Technology, Collaborative Innovation Center of Food Safety and Quality Control in Jiangsu Province, School of Food Science and Technology, Jiangnan University, Wuxi 214122, China; wangqiong9595@163.com (Q.W.); zhanghui@jiangnan.edu.cn (H.Z.); jqzwuxi@163.com (Q.J.)

**Keywords:** linoleic acid, n-6 polyunsaturated fatty acids, blood lipid profiles, dyslipidemia, meta-analysis

## Abstract

Th aim of this meta-analysis was to elucidate whether dietary linoleic acid (LA) supplementation affected blood lipid profiles, including triglycerides (TG), total cholesterol (TC), high-density lipoprotein cholesterol (HDL-C), and low-density lipoprotein cholesterol (LDL-C), compared with other fatty acids. Embase, PubMed, Web of Science and the Cochrane Library databases, updated to December 2022, were searched. The present study employed weighted mean difference (WMD) and a 95% confidence interval (CI) to examine the efficacy of the intervention. Out of the 3700 studies identified, a total of 40 randomized controlled trials (RCTs), comprising 2175 participants, met the eligibility criteria. Compared with the control group, the dietary intake of LA significantly decreased the concentrations of LDL-C (WMD: −3.26 mg/dL, 95% CI: −5.78, −0.74, I^2^ = 68.8%, *p* = 0.01), and HDL-C (WMD: −0.64 mg/dL, 95% CI: −1.23, −0.06, I^2^ = 30.3%, *p* = 0.03). There was no significant change in the TG and TC concentrations. Subgroup analysis showed that the LA intake was significantly reduced in blood lipid profiles compared with saturated fatty acids. The effect of LA on lipids was not found to be dependent on the timing of supplementation. LA supplementation in an excess of 20 g/d could be an effective dose for lowering lipid profiles. The research results provide further evidence that LA intake may play a role in reducing LDL-C and HDL-C, but not TG and TC.

## 1. Introduction

Abundant epidemiologic studies have demonstrated that cardiovascular disease (CVD) is one of the primary underlying causes of mortality worldwide [1]. Dyslipidemia, characterized by elevated levels of triglycerides (TG), total cholesterol (TC), and low-density lipoprotein cholesterol (LDL-C), as well as reduced levels of high-density lipoprotein cholesterol (HDL-C), plays a pivotal role in the onset and progression of CVD [2]. Dyslipidemia is primarily a consequence of the interplay between genetic defects and environmental factors (such as diet, exercise, and drugs effect), in which the type of fatty acids in the diet possibly has a pronounced influence on blood lipid profiles [3].

Linoleic acid (LA) is classified as an essential fatty acid for humans, and the average intake of LA in many countries around the world far exceeds the estimated minimum requirement [4]. LA is widely present in the diet and comes from various sources, such as vegetable oil, nuts, meat, eggs, milk and margarine [5]. There has been increasing evidence that LA has a variety of positive effects on human health, including regulating metabolism, improving insulin resistance, lowering blood lipids and blood pressure, inhibiting cancer cell proliferation and maintaining bone health [6,7,8,9]. However, some studies have found that LA can play a role in promoting inflammation during disease. On the one hand, the downstream product of LA, arachidonic acid (AA), can increase the biosynthesis of proinflammatory eicosane [10]. On the other hand, LA competes with n-3 polyunsaturated fatty acids (PUFA) and inhibits alpha-linolenic acid (ALA), which is converted into docosahexaenoic acid (DHA), because the LA and n-3 PUFAs share the same synthetic pathway in humans.

Based on epidemiological findings, previous studies have confirmed a link between LA supplementation and a reduced risk of CVD and atherosclerosis [11,12,13]. Certain meta-analyses of LA intervention trials have revealed the potential long-term advantages of LA intake regarding its capacity to lower the risk of mortality [14] and type 2 diabetes [15]. Some previous clinical trials have determined that LA might represent a useful tool for designing diets to reduce blood lipids. For instance, researchers who studied the consumption of LA for 45 days significantly attenuated TC and LDL in participants with abdominal adiposity [16]. A pilot study with 50 participants demonstrated that blood concentrations of TG were significantly lower [17]. In addition, results from the intervention studies of LA supplementation on blood lipid profiles were inconsistent, owing to differences in the study design, intervention type, and health status of the participants. Oliveira de Lira found that diets rich in LA resulted in slight increases in TG, TC, and LDL levels in participants [18]. LA intake was associated with a 20% increase in TG and a 10% decrease in HDL-C levels in healthy subjects [19]. However, some studies have shown that the effects of LA intake are neutral on blood lipids [20,21].

A previous systematic review suggested that increasing n-6 fatty acids reduces blood TC by six percent for at least one year, and may make little or no difference to TG, HDL or LDL [22]. The results of the meta-analysis depended on which dietary elements were replaced by n-6 fatty acids in the diet (such as saturated or monounsaturated fatty acid, or other dietary components). This Cochrane review was consistent with the results of Sacks in 2017 [23] in terms of seeing reductions in serum TC when n-6 fatty acids replace saturated fatty acid. This study had certain limitations. The n-6 fatty acids high group contained a certain amount of gamma-linolenic acid (GLA) and trans fatty acids (TFA), which may have confused the effects of LA. As the dominant n-6 PUFA in most diets, the effect of LA on the blood lipids profiles is of great significance to human health. Studies have proven that n-3 PUFA can reduce blood lipids [24,25], but relatively few studies have focused on the effect of replacing ALA with LA on blood lipids. The type of fatty acids that is replaced by LA in the diet may account for the heterogeneity among studies when assessing the effect on blood lipids. Therefore, the purpose of this systematic review and meta-analysis was to elaborately evaluate and compare the effects of the dietary intake of LA when it replaces other types of fatty acids on blood lipids based on randomized controlled trials.

## 2. Materials and Methods

### 2.1. Search Strategy and Selection Studies

This systematic review and meta-analysis were designed and analyzed based on the guidelines of the PRISMA statement. The systematic review was registered in the PROSPERO database under the number CRD42020196824. The Embase, PubMed, Web of Science and the Cochrane Library databases were systematically searched, through to December 2022, using the following keywords in titles and abstracts: (“linoleic acid” OR “linoleate” OR “n-6 fatty acid” OR “soybean oil” OR “peanut oil” OR “walnut oil” OR “sesame oil” OR “safflower oil” OR “sunflower oil” OR “grape seed oil” OR “cottonseed oil” OR “almond oil” OR “corn oil” OR “rice oil”) AND (“lipid” OR “lipemia” OR “lipidemia” OR “hyperlipemia” OR “hyperlipidemia” OR “dyslipidemia” OR “lipoprotein” OR “dyslipoproteinemia” OR “TG” OR “triglyceride” OR “hypertriglyceridemia” OR “TC” OR “cholesterol” OR “hypercholesterolemia” OR “HDL” OR “high density lipoprotein” OR “HDL-Cholesterol” OR “HDL-C” OR “LDL” OR “low-density lipoprotein” OR “LDL-Cholesterol” OR “LDL-C”) AND (“RCT” OR “randomized controlled trial” OR “random” OR “randomly” OR “randomized” OR “parallel” OR “crossover” OR “trial” OR “intervention” OR “placebo” OR “supplement” OR “supplementation”) NOT (“animals”). Moreover, a manual reference check was conducted for the references of the related trials and previous related reviews in order to find further relevant studies.

### 2.2. Eligibility Criteria

Relevant studies that met the following inclusion criteria were selected: (1) randomized parallel/crossover trials only that investigated the effects of linoleic acid on lipid profiles; (2) an intervention duration of at least 4 weeks on individuals aged 18 years or more, with pregnant women excluded; and (3) the intervention group was provided with LA either through their diet or supplements, with only differences in fatty acid intake compared to the control group. Moreover, publications were excluded if they (1) included LA supplementation in combination with other interventions; (2) were animal or in vitro studies, or involved parenteral nutrition; (3) did not provide a sufficient outcome for both the baseline and the end of study; and (4) focused on increased conjugated linoleic acid (CLA), eicosapentaenoic acid (EPA), DHA, or docosapentaenoic acid (DPA) levels.

### 2.3. Data Extraction and Quality Assessment

Data extraction and bias assessment were conducted independently by two researchers (WQ and ZH), in accordance with the Cochrane Handbook of Systematic Reviews. Any discrepancy or disagreement was discussed and resolved by the third investigator (JQ). Relevant data, including the first author’s name and the participant demographics, such as country, age, sex, health status and body mass index (BMI), were extracted from each article. The study design, intervention duration, intervention type and dose, mean change in blood lipid levels between the baseline and endpoint (with standard deviations provided where available) were also recorded. When the effect sizes were measured multiple times at different stages of the trial, the effect size after the longest duration of the intervention was used in the present meta-analysis. The quality of each article was assessed based on the following criteria: (1) random sequence generation; (2) allocation concealment; (3) blinding of participants and personnel; (4) blinding of outcome assessment; (5) incomplete outcome data; (6) selective outcome reporting; and (7) other potential sources of bias. The risk of bias in each item was rated as three categories: low risk, high risk or unclear risk.

### 2.4. Statistical Analysis

The pooled effect size was determined using the mean change and standard deviation (SD) of TC, HDL-C, LDL-C, and TG levels. When the SD was not reported directly, it were estimated using the following formulas: SD = SEM × square root (n); SD = IQR ÷ 1.35; SD = square root (n) × (upperlimit – lowerlimit) ÷ 3.92; SD_change_ = square root [(SD_pre-treatment_)^2^ + (SD_post-treatment_)^2^ – (2R × SD_pre-treatment_ × SD_post-treatment_)] (R = 0.5) [26,27]. All lipid levels were compared at mg/dL, and lipid values reported as mmol/L, divided by 0.0113 (for TG) and 0.0259 (for TC), were converted to mg/dL. Only data in graphical form were digitized and extracted using GetData graphical digitizer software. The pooled effect sizes of the studies were presented as the weighted mean difference (WMD) and 95% confidence interval (CI) for blood lipid levels using the random effects model (DerSimonian-Laird, D-L). The extent of heterogeneity across studies was estimated by using the I^2^ index and the *p*-value obtained from the Q-test. Substantial heterogeneity was defined as an I^2^ value over 50%. In order to investigate the impact of the dietary intake of LA on blood lipids, a pre-planned subgroup analysis was conducted based on age, BMI, supplementation duration, health status, and LA source. To assess internal sensitivity, a leave-one-out analysis was conducted, where one study was omitted from the analysis at a time. Egger’s regression asymmetry test and Begg’s rank correlation test were used to assess the possible publication bias. STATA SE 16.0 (Stata Corporation, College Station, TX, USA) was used to conduct the statistical analyses. *p* < 0.05 was considered statistically significant.

## 3. Results

### 3.1. Literature Search

The flow diagram screening and process of selecting the articles is illustrated in Figure 1. A total of 3700 publications were found after searching the Embase, PubMed, Web of Science, and Cochrane databases and after removing duplicates. After the initial title and abstract screening, 101 full-text articles were assessed for further examination. Subsequently, 61 articles were excluded for the reasons detailed in the flow chart (10 studies had no biomarkers for lipids; the dietary or supplement LA did not meet the set requirements in 11 studies; 7 of the studies had no suitable control group; and 33 articles did not provide sufficient data for meta-analysis). In the end, a total of 40 articles were included in the qualitative synthesis (systematic review).

### 3.2. Study Characteristics

The general characteristics of each study conducted between 1990 and 2022 are illustrated in Table 1. The present analysis included a total of 2175 subjects, with the number of participants in each individual study ranging from 11 to 195. In total, 12 studies included healthy subjects [19,28,29,30,31,32,33,34,35,36,37,38], 9 studies included subjects with hyperlipidemia [16,17,20,39,40,41,42,43,44,45], and other trials were conducted on patients with obesity [17,18,41,46,47,48,49], type 2 diabetes [50,51,52], nonalcoholic fatty liver [53,54], metabolic syndrome [21,55], polycystic ovary syndrome [56,57], hypertension [58], cardiovascular disease [59], chronic peripheral artery occlusive disease [60], peripheral vascular disease [61] or hyperfibrinogenaemia [62]. The design of 30 trials was parallel and 10 studies used a cross-over design. The mean BMI of participants was between approximately 20.2 kg/m^2^ [30] and 34.1 kg/m^2^ [56,57], and the average age was between 21.6 [36] and 72 years old [49]. The selected studies were conducted in the USA (4 trials), Canada (5 trials), Europe (15 trials), Asia (10 trials), Australia (3 trials), Brazil (2 trials), and South Africa (1 trials). In total, 10 studies were conducted in mixed-smoker participants, while 16 studies included only non-smoking participants. Eight trials were conducted only on males, while six trials were performed exclusively on females. The dosage of LA was between 1.36 g/day [33] and 50.75 g/day [31] approximately, and the intervention duration was between 4 and 26 weeks. Different forms of LA consumption were increased through the supplementation of corn oil in 10 trails, safflower oil in 7 trails, soybean oil in 7 trails, and sunflower oil in 16 trails.

### 3.3. Quality Assessment

A summary of the risk-of-bias assessment is provided in Appendix A, using the Cochrane Collaboration tool. The included studies mentioned randomly assigned participants and only 16 trials mentioned a method for generating random sequences. Allocation concealment was rated as low risk in only five studies, with the remaining studies rated as having unclear risk. A double-blind design was substantiated in 23 of the included studies, and a single-blind design was conducted in seven of the included studies. Furthermore, two studies showed a low risk of bias regarding the blinding of the outcome assessments and two studies had a high risk of bias for this parameter. Seven included trials had an unclear risk of bias.

### 3.4. Effect of LA on Blood Lipid Profiles

Among the eligible studies, there was no significant change in TC concentration after LA intake, showing high levels of heterogeneity (WMD: −3.10 mg/dL, 95% CI: −7.52, 1.32, *p* = 0.17, I^2^ = 80.8%). As shown in Table 2 and Figure 2, the subgroup analysis indicated a higher significant reduction in TC using studies employing LA supplementation compared with SFA and MUFA supplementation, but not for n-3 PUFA supplementation. Furthermore, when the studies were grouped by BMI, the significant effects of LA on TC levels were evidenced in BMI < 30 kg/m^2^ (WMD: −5.88 mg/dL, 95% CI: −10.04, −1.73, *p* < 0.01, I^2^ = 61.8%).

As shown in Table 2 and Figure 3, the overall results from the random effects model revealed that LA consumption increased HDL-C significantly less than in the control groups (WMD: −0.64 mg/dL, 95% CI: −1.23, −0.06, *p* = 0.03), with moderate levels of heterogeneity among the studies (I^2^ = 30.3%, *p*_for heterogeneity_ = 0.03). Considering the control groups with different types of fatty acids, LA intake had pronounced effects on the HDL-C concentration compared to SFA intake (WMD: −0.95 mg/dL, 95% CI: −1.63, −0.27, *p* < 0.01, I^2^ = 0.0%; Table 2 and Figure 3). In addition, HDL-C increased significantly less after LA intake in the subgroups of BMI ≥ 30 kg/m^2^ (WMD: −1.39 mg/dL, 95% CI: −2.74, −0.04, *p* = 0.04, I^2^ = 75.2%). With doses of 0–10 g/d and a duration of <12 weeks, the beneficial effect of LA was attenuated regarding HDL-C levels.

In comparison to the control group, the pooled results demonstrated that LA supplementation led to a noteworthy reduction in blood LDL-C concentration by 3.26 mg/dL (95%CI: −5.78, −0.74, *p* = 0.01). A moderate level of heterogeneity was observed across studies (I^2^ = 68.8%, *p*_for heterogeneity_ = 0.00). As shown in Table 2 and Figure 4, LA supplementation decreased LDL-C by 4.19 mg/dL in the subgroup of ≤ 50 years of age (95% CI: −7.19, −1.18, *p* < 0.01, I^2^ = 75.8%). The subgroup analysis found that LA supplementation resulted in a greater reduction in blood LDL-C than that found in the SFA group (WMD: −7.65 mg/dL, 95% CI: −11.79, −3.52, *p* < 0.01, I^2^ = 73.6%). LDL-C levels were significantly reduced by −3.74 in subjects with normal lipids (95% CI: −5.46, −2.02, *p* < 0.01, I^2^ = 0.0%), 4.11 in those with a BMI of < 30 kg/m^2^ (95% CI: −6.60, −1.61, *p* < 0.01, I^2^ = 54.5%), and 2.56 in those supplemented with sunflower oil (95% CI: −5.20, −0.09, *p* = 0.04, I^2^ = 15.8%). Interestingly, LA supplementation at doses greater than 20 g/d induced a significant decrease in TC and LDL-C levels, while doses of 0–20 g/d had no significant effect.

The pooled results obtained using the random effects model indicated that LA consumption had a non-significant effect on the serum TG level compared to other fatty acids (WMD: 1.83 mg/dL, 95% CI: −2.91, 6.56, *p* = 0.45). There was significant heterogeneity across the trials (I^2^ = 74.0%, *p*_for heterogeneity_ = 0.00). In Table 2 and Figure 5, a subgroup analysis was conducted based on different types of control groups, revealing that the LA supplementation group exhibited a significant reduction in TG levels compared to the SFA consumption group (WMD: −3.33 mg/dL, 95% CI: −5.99, −0.68, *p* = 0.01, I^2^ = 0.0%). Furthermore, the reduction in blood TG was significantly lower for the LA consumption group compared to the n-3 PUFA consumption group (WMD: 9.42 mg/dL, 95% CI: 1.40, 17.44, *p* = 0.02, I^2^ = 44.7%). After categorizing the studies based on the type of intervention, TG was significantly decreased for LA supplementation using corn oil. Meanwhile, LA consumption using soybean oil had reduced serum TG significantly less. In terms of dose difference, the overall trend was that the effect of LA intake decreased with dose, and that significant differences were found regarding the ratio of TG in the studies stratified by dose (*p* for meta regression =0.027) using meta-regression analysis (Figure 6).

As summarized in Figure 6, visual inspection of the funnel plots did not confirm any evidence of publication bias with the effect of OA supplementation on CRP (Begg’s test: *p* = 0.078, Egger’s test: *p* = 0.174), TNF (Begg’s test: *p* = 0.732, Egger’s test: *p* = 0.952), IL-6 (Begg’s test *p* = 0.692, Egger’s test: *p* = 0.878), or sICAM-1 (Begg’s test *p* = 0.368, Egger’s test: *p* = 0.833).

### 3.5. Publication Bias and Sensitivity Analysis

There was no sign of publication bias in the meta-analysis of LA supplementation on TC, HDL-C, and LDL-C through visual inspection of funnel plot (Appendix A). However, Egger’s linear regression tests and visual inspection of funnel plot revealed a publication bias for TG (*p* = 0.002, Egger’s test and *p* = 0.142, Begg’s test). There were no missing studies were evaluated for the effect of LA consumption on TG by the “trim and fill” method. Sensitivity analysis performed that no study had a significant effect on the overall effect sizes of blood lipids profiles (Appendix A).

## 4. Discussion

Dyslipidemia, particularly higher concentrations of TC and LDL-C, is significantly correlated with an elevated risk of CVD. Several effective food nutrients have been developed to improve lipid profiles, including the use of fatty acids [63,64]. LA is one of the essential fatty acids and is mainly found in corn oil, sunflower oil, soybean oil, safflower oil and peanut oil. It has been reported that increasing LA consumption exerts no influence on inflammatory markers [65].

Previous meta-analyses have shown that increasing n-6 fat can lower the blood TC without affecting TG, HDL, or LDL [19]. However, there are some limitations to this study. The studies reviewed provided few details about the type of n-6 fatty acids the participants consumed to increase their n-6 fatty acids intake. The results of the meta-analysis resulted in combined data on GLA and LA, which may not be appropriate. LA is an essential fatty acid that is available from a variety of dietary fats and oils, while GLA had to be provided in the trial via supplement capsules, so their effects may differ. Another key question relates to TFA intake. Some trials may have increased the TFA intake in the high n-6 fatty acids group, which could confuse our understanding of the effects of increasing n-6 fatty acids. This study has provided a systematic review and meta-analysis that evaluates the effects of dietary LA intake and its ability to replace other types of fatty acids on blood lipids profiles. With the inclusion of 40 randomized controlled trials comprising 2175 participants, the analysis revealed a significant reduction in LDL-C and HDL-C concentrations with LA intake compared to other fatty acids. However, the research results did not indicate a significant effect of LA on TG and TC levels.

The subgroup analysis showed that LA lowered TG, TC and LDL-C levels more than SFA. Most studies demonstrated that the replacement of SFAs with PUFAs and MUFAs could reduce lipid profiles [66,67,68]. It has been proposed that ALA and OA could significantly reduce blood lipids profiles, whereas OA is not able as ALA to lower TG and TC [50,69,70]. Moreover, Aguilera et al. concluded that LA decreased TC more effectively than MUFA in subjects with abdominal obesity [47,60]. Meta-analysis pointed out that MUFA reduced TC less than n-6 fatty acids [71]. Similarly, the results of this meta-analysis and subgroup analysis demonstrated that LA decreased the TC level more than MUFA. Squalene is a hydrocarbon that has long been considered to have hypercholesterolemic properties [72]. Olive oil is the main source of MUFA and has a higher content of squalene than n-6 PUFA-based vegetable oil. Thus, it can be inferred that MUFA is not as effective as LA in reducing TC.

This study did not show that LA significantly reduced TG, and the n-3 polyunsaturated fatty acid intake significantly decreased TG more than LA according to the subgroup analysis. Yue’s study published in 2020 found that ALA intake significantly changes the concentrations of TG, TC and LDL-C [73]. LA may exert an influence on lipid regulation by affecting the key enzymes and proteins in lipid synthesis. On the other hand, the consumption of n-3 PUFA could reduce TG accumulation, both by reducing the expression of the key enzyme fatty acid synthase (FAS) and of HMG-CoA-Hydroxy-3-Methylglutaryl CoA Reductase (HMG-CoAR) in lipid synthesis, and by increasing the expression of the peroxisome proliferator-activated receptor (PPAR), a key enzyme that promotes lipid metabolism and fatty acid oxidation [74]. In addition, the inhibition of the sterol regulatory element binding protein (SREBP), apo-B100, and very low-density lipoprotein cholesterol (VLDL-C) synthesis could thereby reduce blood TG and TC [75].

The current study showed that LA increases HDL-C less compared to the other control groups. Some studies reported that a high PUFA intake is associated with lower HDL-C levels compared to high SFA and MUFA intake [18,32,48]. Similar results were revealed by Ghobadi, showing that OA has a greater beneficial effect on HDL-C compared to LA [71]. The subgroup analysis also found that LA intake increased HDL-C less than SFA, which was consistent with the above evidence. This discrepancy may be due to the distinct mechanism of HDL-C when compared to other lipid indicators. Furthermore, soybean oil significantly decreased HDL-C and increased TG compared with other vegetable oils. The subgroup analysis showed that corn oil lowered TG and sunflower oil lowered LDL-C more than other vegetable oils. Different types and contents of sterol, polyphenols, and tocotrienol compounds in vegetable oils can reduce lipid levels in different ways. Plant sterols reduce cholesterol esterification and inhibit the intestinal absorption of cholesterol due to their structural similarity with cholesterol [76]. Polyphenols can lower lipids and inhibit accelerated atherosclerosis by stimulating AMP-activated protein kinase [77]. Tocotrienol inhibits the accelerated oxidation of cholesterol via amidine dihydrochloride 2-methylpropionate [78].

The decreasing effect of 20 g/d of LA on TC and LDL-C was obviously stronger than that of lower doses of LA. When the LA intake was less than 20 g/d, the health effects of low doses were not significant. This may mean that LA supplementation in excess of 20 g/d may be an effective dose for lowering lipid profiles. At the same time, the addition of 0–10 g/d of LA significantly reduced the HDL-C concentration, which was consistent with the previous hypothesis.

Subgroup analyses indicated that LA supplementation resulted in significantly lower TC levels than the control group in subjects with a BMI of less than 30 kg/m^2^. For LDL-C, the effect of LA was more evident among people that were less than 50 years old and had a BMI of less than 30 kg/m^2^. This meta-analysis showed that, for TC and LDL-C, the reducing effect is diminished as the BMI and age increase. Elevated oxidative stress in the very elderly can lead to reduced levels of both serum TG and LDL-C, as well as reduced levels of antioxidants [66]. Older adults taking drugs may see a change in their serum lipids; therefore, the effect of fatty acids was not evident. Furthermore, the impact of LA supplementation on LDL-C was only significant in patients with normolipidemia. It is speculated that this may be due to use of medication in hyperlipidemic patients, as well as other pathological causes. For instance, lipid profiles were affected by insulin resistance in patients with metabolic syndrome [72,79,80]. TC concentrations were lower in subjects with active rheumatoid arthritis [81]. Therefore, future intervention studies will require more precise randomized clinical trials.

LA may regulate blood lipid levels via multiple mechanisms. The intake of LA in the diet increased the hepatic expression of the LDL receptor (LDLR), thereby reducing liver adipogenesis [82]. It was found that LA mediated lipid catabolism by inhibiting the activity of SREBP-1c [83]. The LA-rich diet reduced the levels of proprotein convertase subtilisin/kexin type 9 (PCSK9) in the blood, increased the amount of LDLR on liver cells, and thus improved the clearance rate of LDL in the blood [30]. Replacing SFA with LA reduces the production and quantity of LDL particles by reducing the synthesis of apolipoprotein B10 [84]. There are several strengths of this study. The present study systematically demonstrated the relationship between LA and lipid profiles for the first time, that is, LA had significant reduced HDL-C and LDL-C concentrations. The study had a relatively high statistical power, since it involved 40 independent RCTs and 2175 subjects in 21 countries. Simultaneously, the meta-analysis has several limitations. The subgroup analysis showed that the duration of the study exerted an influence on the overall results, with all the included trials being conducted for relatively short periods (≤6 months). While this study demonstrated that the duration exerted an influence on the relationship between LA intake and blood lipids, all the included trials had a shorter duration (≤24 weeks). Accordingly, the long-term effects of LA need to be explored in the future. LA comes from a variety of vegetable oils, with different fatty acid compositions and other bioactive ingredients. Changes in other minor components may confuse the impact of LA on lipid levels. The consumption of energy and the composition of the macronutrients were not constant during the intervention period in all the research. Whether the role of LA is affected by the above factors remains to be further studied. Finally, the influence of other potential factors (genetic differences and lifestyle changes) could not be evaluated.

## 5. Conclusions

In conclusion, this systematic review is the first to comprehensively synthesize the results of LA regulation on lipids from randomized controlled trials. This meta-analysis suggested that a dietary intake of LA significantly decreased blood LDL-C and HDL-C concentrations. The subgroup analysis provided evidence that a dietary intake of LA can significantly decrease blood lipid profiles compared to the SFA. LA supplementation in excess of 20 g/d may be an effective dose for lowering lipid profiles. The effect of LA was more evident among young healthy people and those with a BMI of less than 30 kg/m^2^.

## Figures and Tables

**Figure 1 foods-12-02129-f001:**
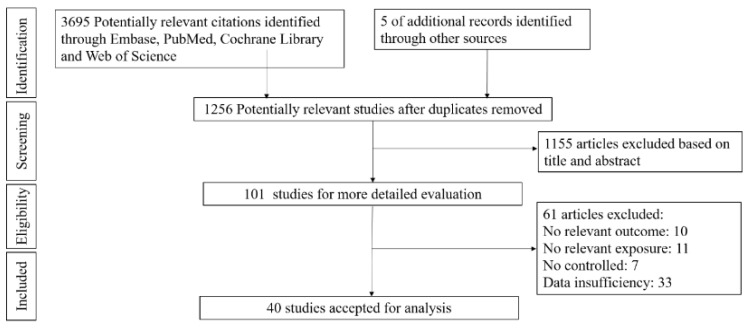
Flow chart of study selection.

**Figure 2 foods-12-02129-f002:**
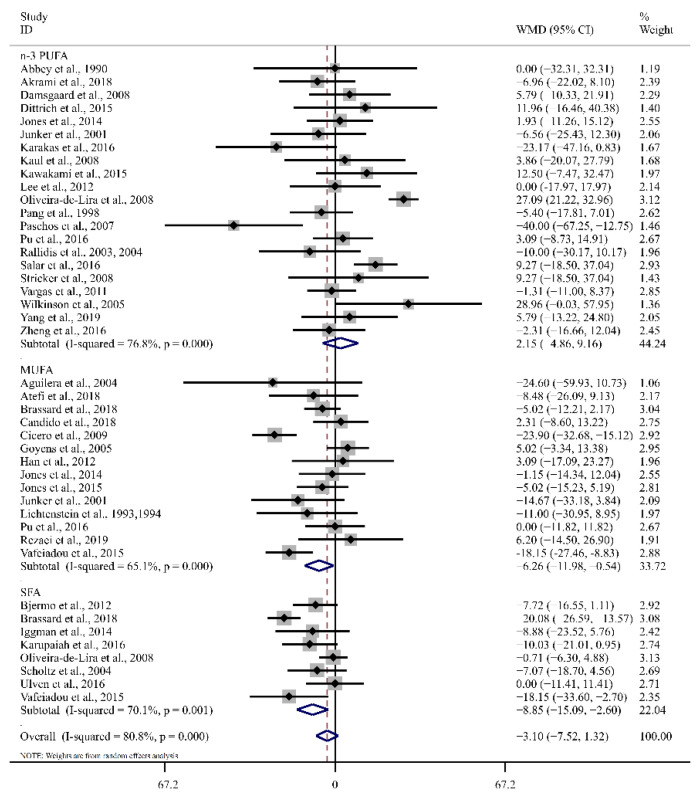
Forest plot of the effect of LA supplementation on TC. Ref. [16,17,18,19,20,21,28,29,30,31,32,33,34,35,37,38,39,40,41,42,43,44,45,46,47,48,50,51,52,54,55,56,57,58,59,60,61,62]. The diamond represents the overall effect estimate of the meta-analysis and the black point represents mean difference of effect measure of each study.

**Figure 3 foods-12-02129-f003:**
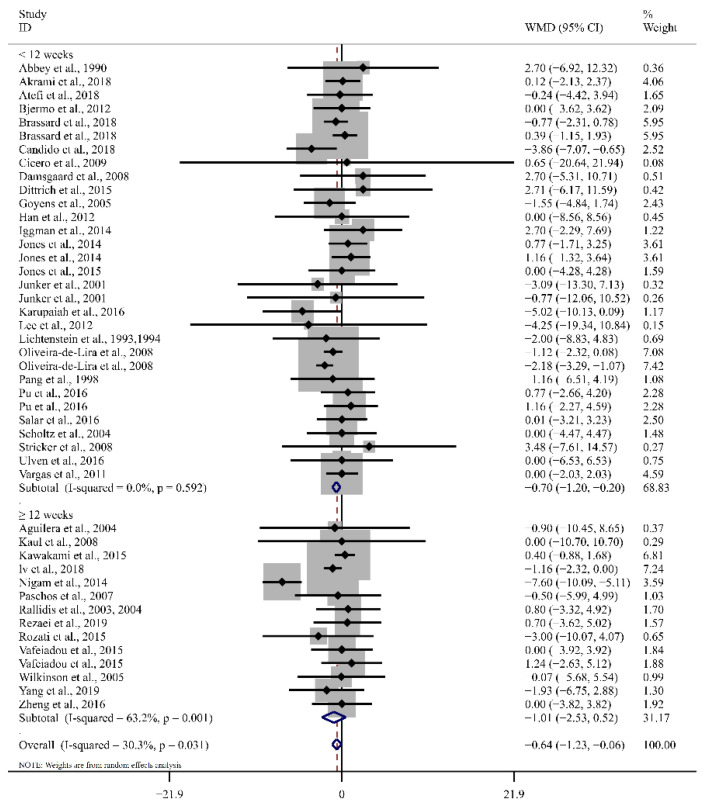
Forest plot of the effect of LA supplementation on HDL-C. Ref. [16,17,18,19,20,21,28,29,30,31,32,33,34,35,36,37,38,39,40,41,42,43,44,45,46,47,48,49,50,51,52,53,54,55,57,58,59,60,61,62]. The diamond represents the overall effect estimate of the meta-analysis and the black point represents mean difference of effect measure of each study.

**Figure 4 foods-12-02129-f004:**
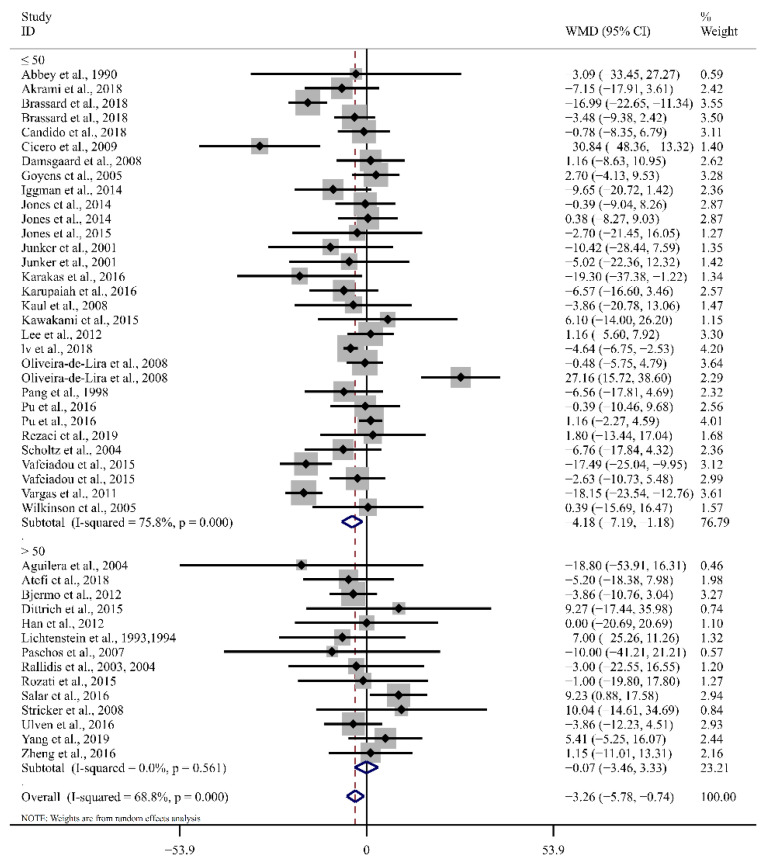
Forest plot of the effect of LA supplementation on LDL-C. Ref. [16,17,18,19,20,21,28,29,30,31,32,33,34,35,36,37,38,39,40,41,42,43,44,45,46,47,48,49,50,51,52,53,54,55,56,57,58,59,60,61,62]. The diamond represents the overall effect estimate of the meta-analysis and the black point represents mean difference of effect measure of each study.

**Figure 5 foods-12-02129-f005:**
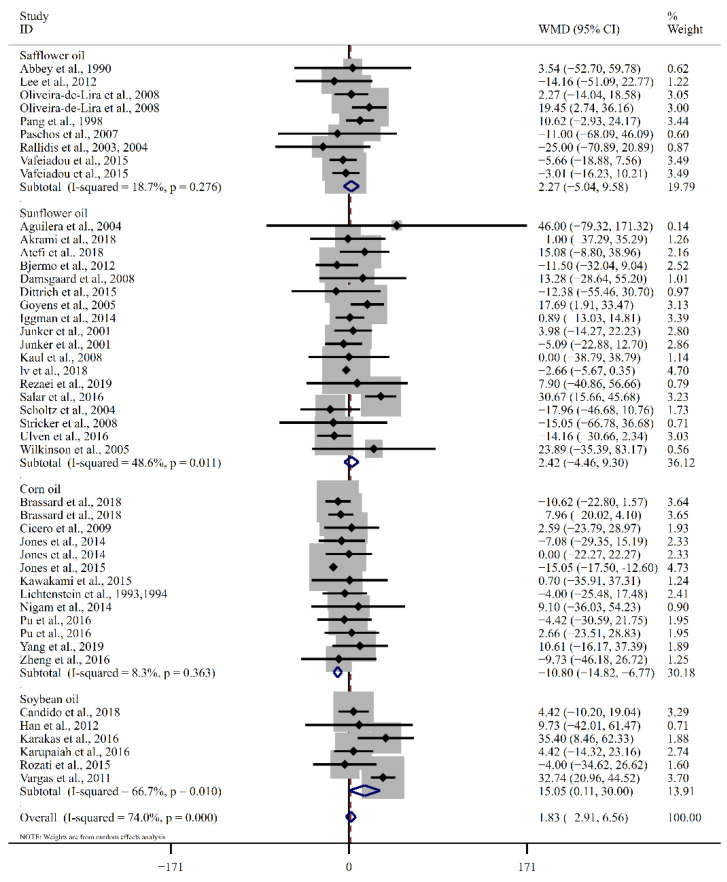
Forest plot of the effect of LA supplementation on TG Ref. [16,17,18,19,20,21,28,29,30,31,32,33,34,35,36,37,38,39,40,41,42,43,44,45,46,47,48,49,50,51,52,53,54,55,56,57,58,59,60,61,62]. The diamond represents the overall effect estimate of the meta-analysis and the black point represents mean difference of effect measure of each study.

**Figure 6 foods-12-02129-f006:**
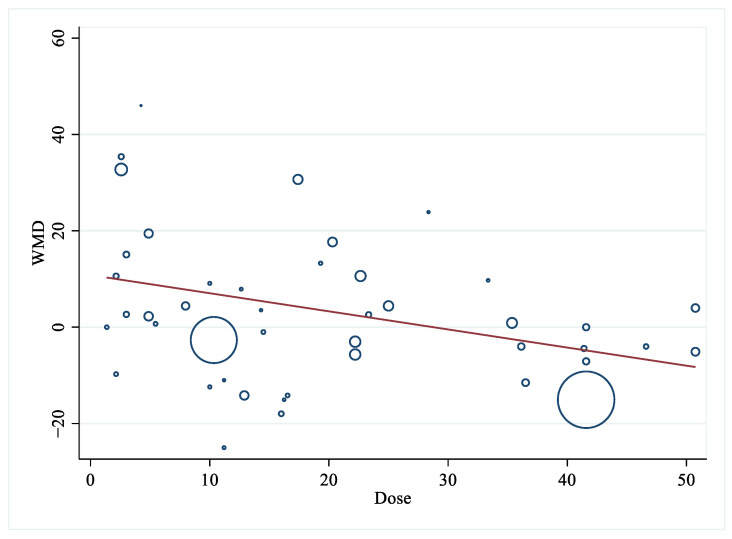
Meta-regression for dose and effect size (LA supplementation on TG).

**Table 1 foods-12-02129-t001:** Characteristics of 40 randomized controlled trials selected for meta-analysis.

Reference	Country	Subjects Information	Age	BMI	Smoking	No.	M/F	Duration	Design	LA Dose (Source)
Abbey et al. (1990) [39]	Australia	Hypercholesterolaemia	47.4	26.1	NR	22	22/0	6 weeks	P	I: 14.3 g (Safflower oi) C: 4.3 g (Linseed oil)
Aguilera et al. (2004) [61]	Spain	Peripheral vascular disease	65.1	26.7	Mixed	20	20/0	16 weeks	P	I: 4.23 g (Sunflower oil) C: 0.52 g (Olive oil)
Akrami et al. (2018) [55]	Iran	Metabolic syndrome	48.6	NR	Non-smoker	52	33/19	7 weeks	P	I: 14.5 g (Sunflower oil) C: 4.03 g (Flaxseed oil)
Atefi et al. (2018) [50]	Iran	Type 2 diabetes mellitus	58	28.5	Non-smoker	81	0/81	8 weeks	P	I: 3 g (Sunflower oil) C: 0 g (Olive oil)
Bjermo et al. (2012) [46]	Sweden	Abdominal obesity	56.5	30.8	NR	61	NR	10 weeks	P	I: 36.5 g (Sunflower oil) C: 0 g (Butter)
Brassard et al. (2018) [47]	Canada	Abdominal obesity	41	29.9	Non-smoker	46	21/25	4 weeks	CO	I: NR (Corn oil) C: NR (Butter) C: NR (Olive oil)
Candido et al. (2018) [48]	Brazil	Overweight or obese	27	30.1	Non-smoker	41	0/41	9 weeks	P	I: 25 mL (Soybean oil) C: 25 mL (Olive oil)
Cicero et al. (2009) [16]	Italy	Moderate hypercholesterolaemia	50	26.2	NR	22	11/11	45 days	P	I: 23.33 g (Corn oil) C: 9.05 g (Olive oil)
Damsgaard et al. (2008) [28]	Denmark	Healthy	25	23.3	Mixed	33	33/0	8 weeks	P	I: 19.3 g (Sunflower oil) C: 12.3 g (Rapeseed oil)
Dittrich et al. (2015) [40]	Germany	Moderate hypertriacylglyceridemia	56	28.2	NR	49	17/32	10 weeks	CO	I: 10 g (Sunflower oil) C: 3.1 g (Linseed oil)
Goyens et al. (2005) [29]	Netherlands	Healthy	49.6	24.1	Mixed	36	14/22	6 weeks	P	I: 3.15 g (Sunflower oil, olive oil and rapeseed oil) C: 1.1 g (Olive and rapeseed oil)
Han et al. (2012) [20]	Korea	Moderately hypercholesterolemia	63	26.7	Non-smoker	18	7/11	35 days	CO	I: 33.36 g (Soybean oil) C: 5.1 g (high oleic acid soybean oil)
Iggman et al. (2014) [30]	Sweden	Healthy	26.9	20.2	NR	39	12/27	7 weeks	P	I: 35.36 g (Sunflower oil) C: 4.9 g (Palm oil)
Jones et al. (2014) [41]	Canada	Abdominal obesity (some with hyperlipidemia)	46.5	29.8	Non-smoker	130	60/70	4 weeks	CO	I: 41.58 g (Corn and safflower oil) C: 8.82 g (High oleic canola oil) C: 22.5 g (Flaxseed oil)
Jones et al. (2015) [17]	Canada	Abdominal obesity (some with hyperlipidemia)	45.8	30.4	NR	50	26/24	4 weeks	CO	I: 41.58 g (Corn and safflower oil) C: 8.82 g (High oleic canola oil)
Junker et al. (2001) [31]	Germany	Healthy	26	23	Non-smoker	58	31/27	4 weeks	P	I: 50.75 g (Sunflower oil) C: 7.76 g (Olive oil) C: 18.35 g (Rapeseed oil)
Karakas et al. (2016) [56]	USA	Polycystic ovary syndrome	29.2	34.1	Non-smoker	34	0/34	6 weeks	P	I: 2.57 g (Soybean oil) C: 0.97 g (Flaxseed oil)
Karupaiah et al. (2016) [32]	Malaysia	Healthy	23.4	25.1	NR	34	16/18	4 weeks	CO	I: 7.97 g (Soybean oil) C: 1.85 g (Palm oil)
Kaul et al. (2008) [33]	Canada	Healthy	34	24.3	Non-smoker	44	17/27	12 weeks	P	I: 1.36 g (Sunflower oil) C: 0.28 g (Flaxseed oil)
Kawakami et al. (2015) [34]	Japan	Healthy	44.5	25.1	Mixed	15	15/0	12 weeks	CO	I: 5.45 g (Corn oil) C: 1.62 g (Flaxseed oil)
Lee et al. (2012) [42]	Australia	Hypercholesterolaemia	47	24.9	Non-smoker	11	6/5	4 weeks	CO	I: 16.53 g (Safflower oil) C: 9.3 g (Canola oil)
Lichtenstein et al. (1993) [35]	USA	Healthy	61	27.4	Non-smoker	15	7/8	32 days	CO	I: 36.14 g (Corn oil) C: 6.7 g (Olive oil)
Lv et al. (1993) [36]	China	Healthy	21.6	21	Non-smoker	108	50/58	16 weeks	P	I: 10.34 g (Soybean oil) C: 2.9 g (Palm oil)
Nigam et al. (2014) [53]	India	Nonalcoholic fatty liver disease	36.7	27.3	Mixed	60	60/0	24 weeks	P	I: 20 mL (Corn oil) C: 20 mL (Olive oil)
Oliveira-de-Lira et al. (2018) [18]	Brazil	Abdominal adiposity	34.1	33.9	Non-smoker	75	0/75	8 weeks	P	I: 4.87 g (Safflower oil) C: 0.13 g (Coconut oil)
										C: 0.6 g (Chia oil)
Pang et al. (1998) [19]	Australia	Healthy	24.5	22.4	NR	29	29/0	6 weeks	P	I: 22.65 g (Safflower oil) C: 13.18 g (Linseed oil)
Paschos et al. (2007) [43]	Greece	Nondiabetic dyslipidemia	52	28	Non-smoker	35	35/0	12 weeks	P	I: 11.2 g (Safflower oil) C: 2.07 g (Flaxseed oil)
Pu et al. (2016) [21]	Canada	Metabolic syndrome	45.6	29.6	NR	84	35/49	30 days	CO	I: 41.4 g (Corn and safflower oil) C: 9 g (High oleic canola oil) C: 22.9 g (Flaxseed oil)
Rallidis et al. (2003, 2004) [44,45]	Greece	Dyslipidaemia	51	28.4	Mixed	76	76/0	12 weeks	P	I: 11.2 g (Safflower oil) C: 2 g (Linseed oil)
Rezaei et al. (2019) [54]	Iran	Non-alcoholic fatty liver disease	43.6	30.1	Mixed	66	29/37	12 weeks	P	I: 12.64 g (Sunflower oil) C: 3.08 g (Olive oil)
Rozati et al. (2015) [49]	USA	Overweight or obese	72	29	Non-smoker	41	14/27	12 weeks	P	I: 46.61 g (10% corn oil and 90% soybean oil) C: 8.6 g (Olive oil)
Salar et al. (2006) [51]	Iran	Type 2 diabetes mellitus	52.1	30.2	Non-smoker	99	58/41	8 weeks	P	I: 17.4 g (Sunflower oil) C: 6.39 g (Canola oil)
Scholtz et al. (2004) [62]	South Africa	Hyperfibrinogenaemia	48.1	28.7	Mixed	56	36/20	4 weeks	P	I: 16 g (Sunflower oil) C: 3.2 g (Red palm oil)
Stricker et al. (2008) [60]	Switzerland	Chronic peripheral artery occlusive disease	65	NR	Mixed	40	27/13	8 weeks	P	I: 16.24 g (Sunflower oil) C: 4.5 g (Canola oil)
Ulven et al. (2016) [37]	Norway	Healthy	54.4	25	Mixed	99	58/41	8 weeks	P	I: 12.9 g (Sunflower oil) C: 4.1 g (Butter)
Vafeiadou et al. (2015) [59]	UK	Cardiovascular disease	44	26.7	Non-smoker	195	85/110	16 weeks	P	I: 22.2 g (Safflower oil) C: 7.3 g (Butter) C: 10.1 g (Olive and rapeseed oil)
Vargas et al. (2011) [57]	USA	Polycystic ovary syndrome	29.2	34.1	Non-smoker	34	0/34	6 weeks	P	I: 2.57 g (Soybean oil) C: 10.1 g (Flaxseed oil)
Wilkinson et al. (2005) [38]	UK	Healthy	49	28.3	Non-smoker	38	NR	12 weeks	P	I: 28.35 g (Sunflower oil) C: NR (Flaxseed oil)
Yang et al. (2019) [58]	China	Hypertension	57.5	26.8	NR	73	27/46	12 weeks	P	I: 2.14 g (Corn oil) C: 0 g (Flaxseed oil)
Zheng et al. (2016) [52]	China	Type 2 diabetes mellitus	59.4	25.1	NR	108	35/73	180 days	P	I: 2.14 g (Corn oil) C: 0.62 g (Flaxseed oil)

Abbreviations: BMI, body mass index; F, female; M, male; NR, not reported; No., number of included participants; I, intervention; C, control; P, parallel; CO, crossover. Studies had low bias according to the incomplete outcome data. Selective outcome reporting was rated as an unclear risk of bias in all studies except for 20, which were deemed to have a low risk of bias.

**Table 2 foods-12-02129-t002:** Subgroup analyses for the impact of LA supplementation on blood lipids.

	TG				TC			
Subgroup	N	WMD (95% CI)	*p*	I^2^ %	N	WMD (95% CI)	*p*	I^2^ %
Overall	46	1.83 (−2.91, 6.57)	0.45	74.0	43	−3.10 (−7.52, 1.32)	0.17	80.8
Age								
≤50	32	2.14 (−3.14, 7.43)	0.43	78.2	30	−3.13 (−8.50, 2.24)	0.25	84.5
>50	14	0.24 (−10.77, 11.25)	0.97	47.2	13	−2.57 (−10.09, 4.95)	0.50	61.0
Duration								
<12 weeks	32	2.74 (−3.94, 9.42)	0.42	80.5	32	−2.53 (−7.49, 2.44)	0.32	83.2
≥12 weeks	14	−2.63 (−5.42, 0.17)	0.07	0.0	11	−5.22 (−14.89, 4.45)	0.29	64.4
Intervention groups								
Safflower oil	9	2.27 (−5.05, 9.58)	0.54	18.7	9	−6.13 (−19.98, 7.71)	0.39	92.5
Sunflower oil	18	2.42 (−4.46, 9.30)	0.49	48.6	17	−0.17 (−5.52, 5.18)	0.95	50.6
Corn oil	13	−10.80 (−14.82, −6.77)	<0.01	8.3	12	−5.03 (−11.51, 1.45)	0.13	73.7
Soybean oil	6	15.05 (0.11, 30.00)	0.05	66.7	5	−3.84 (−10.86, 3.19)	0.28	29.0
Main fatty acid in comparison groups								
n-3 PUFA	21	9.42 (1.40, 17.44)	0.02	44.7	21	2.15 (−4.86, 9.16)	0.55	76.8
MUFA	16	−0.38 (−7.86, 7.09)	0.92	62.4	14	−6.26 (−11.98, −0.54)	0.03	65.1
SFA	9	−3.33 (−5.99, −0.68)	0.01	0.0	8	−8.85 (−15.09, −2.61)	<0.01	70.1
Health status								
Normolipemic	14	−0.11 (−4.14, 3.93)	0.96	9.0	13	−1.51 (−6.44, 3.41)	0.55	26.4
Hyperlipidemia	6	−3.9 (−20.93, 13.13)	0.65	0.0	6	−11.25 (−25.59, 3.09)	0.12	65.4
Other disease	26	2.78 (−4.94, 10.50)	0.48	81.6	24	−2.61 (−8.63, 3.42)	0.40	86.3
BMI								
<30 kg/m^2^	32	−2.36 (−4.71, −0.02)	0.05	0.0	30	−5.88 (−10.04, −1.73)	<0.01	61.8
≥30 kg/m^2^	12	10.98 (−3.95, 25.91)	0.15	91.5	11	2.52 (−6.45, 11.49)	0.58	88.3
Dose difference								
0–10 g/d	15	9.08 (−0.54, 18.69)	0.06	55.4	14	0.81 (−7.59, 9.22)	0.85	83.7
10–20 g/d	12	3.20 (−8.44, 14.84)	0.59	60.1	11	−0.01 (−7.60, 7.58)	0.99	60.1
>20 g/d	17	−2.61 (−8.61, 3.40)	0.40	56.3	16	−6.96 (−12.07, −1.86)	<0.01	58.0
Overall	45	−0.64 (−1.23, −0.06)	0.03	30.3	45	−3.26 (−5.78, −0.74)	0.01	68.8
Age								
≤50	31	−0.70 (−1.43, 0.04)	<0.01	49.9	30	−4.19 (−7.19, −1.18)	<0.01	75.8
>50	14	−0.21 (−1.57, 1.14)	0.99	0.0	14	−0.07 (−3.46, 3.33)	0.97	0.0
Duration								
<12 weeks	31	−0.70 (−1.20, −0.20)	<0.01	0.0	32	−3.27 (−6.55, 0.01)	0.05	74.3
≥12 weeks	14	−1.01 (−2.53, 0.52)	0.20	63.2	13	−3.83 (−7.82, 0.17)	0.06	36.0
Intervention groups								
Safflower oil	9	−1.39 (−2.14, −0.64)	<0.01	0.0	9	−1.11 (−9.12, 6.90)	0.79	81.5
Sunflower oil	18	−0.52 (−1.32, 0.29)	0.21	0.0	18	−2.56 (−5.20, −0.09)	0.04	15.8
Corn oil	13	−0.55 (−1.93, 0.84)	0.44	69.2	12	−3.55 (−8.75, 1.65)	0.18	74.9
Soybean oil	5	−2.14 (−4.49, −0.22)	0.08	36.5	6	−8.32 (−16.78, 0.15)	0.05	71.4
Main fatty acid in comparison groups								
n-3 PUFA	20	−0.48 (−1.10, 0.15)	0.14	0.0	21	−0.10 (−4.95, 4.76)	0.97	75.1
MUFA	16	−1.18 (−2.82, 0.47)	0.16	61.5	15	−2.41 (−5.38, 0.53)	0.11	8.6
SFA	9	−0.95 (−1.63, −0.27)	<0.01	0.0	9	−7.65 (−11.79, −3.52)	<0.01	73.6
Health status								
Normolipemic	14	−0.56 (−1.34, 0.21)	0.15	0.0	14	−3.74 (−5.46, −2.02)	<0.01	0.0
Hyperlipidemia	6	0.73 (−2.03, 3.48)	0.61	0.0	6	−7.71 (−20.51, 5.08)	0.24	44.5
BMI								
<30 kg/m^2^	32	−0.30 (−0.89, 0.30)	0.33	0.0	32	−4.11 (−6.60, −1.61)	<0.01	54.5
≥30 kg/m^2^	11	−1.39 (−2.74, −0.04)	0.04	75.2	11	−0.56 (−8.12, 7.01)	0.89	85.9
Dose difference								
0–10 g/d	14	−1.37 (−2.72, −0.02)	0.05	68.4	14	−1.45 (−7.51, 4.6)	0.64	81.9
10–20 g/d	12	−0.61 (−1.47, 0.25)	0.16	0.0	12	−0.58 (−4.11, 2.96)	0.75	39.0
>20 g/d	17	0.02 (−1.01, 1.04)	0.98	0.0	17	−5.25 (−8.79, −1.72)	<0.01	38.3

Abbreviations: CI, confidential interval; SFA, saturated fatty acid; MUFA, monounsaturated fatty acid; n-3 PUFA, n-3 polyunsaturated fatty acid; N, number of included studies.

## Data Availability

Data is contained within the article and Appendix A.

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
