# Peer review of "Effects of Dietary Linoleic Acid on Blood Lipid Profiles: A Systematic Review and Meta-Analysis of 40 Randomized Controlled Trials"

_foods, 2023, doi:10.3390/foods12112129_

Round 1

Reviewer 1 Report

The authors made a systematic review about the effect of dietary LA supplementation on blood lipids. It is an interesting topic, however, some major concerns have been raised during the review process:

1.)    Based on the Introduction it is not clear, why LA supplementation might be beneficial. You wrote, that LA can promote the biosynthesis of pro-inflammatory eicosanoids from AA, and inhibit the conversion of ALA to DHA (so lower the biosynthesis from the n-3 PUFA-derived anti-inflammatory cytokines). Why should the dietary intake of LA be increased despite this? Please clarify it and when necessary, add some relevant articles that prove your aim of the review.

2.)    My other big concern with the methods part, that it is not clear, how you have chosen the control and intervention groups from the included studies. It is quite obvious for studies comparing the results from only two supplementation groups, but in the others comparing the effect of more than two supplementation groups how did you choose the two included groups: e.g. Brassard et al (ref. 38) investigated the effect of SFA-rich diet (in form of either cheese or as butter) MUFA-rich diet (as olive oil) and PUFA-rich diet (in form of corn oil). Why did you choose MUFA-group as controls and not SFA-group? Please clarify it! I suppose, multiple comparisons would be more proper than selecting only a single one because this leads to bias in your publication. So in this case you should compare the effect of corn oil (LA) supplementation with the effect of olive oil (MUFA) AND with butter (SFA) in a pairwise comparison.
Moreover, in Table 1 you state, that intervention group received canola oil, but they received corn oil. Canola oil is a kind of rapeseed oil low in erucic acid, so it is quite high in oleic acid (MUFA) accounting for about 60% of total fatty acids. In contrast, corn oil is rich in LA (about 55% of total fatty acids). In the original study (Brassard et al, 10.3945/ajcn.116.150300) smoking was also part of exclusion criteria, so you can say that they were non-smokers. Please correct it!

3.)    Page 2, line 48: the cited reference (Jones et al, Am J Clin Nutr, 2014) is not a meta-analysis, but a RCT and is not about lowering the risk of death, but mainly blood lipids. The other reference (Zhao et al, BMC Medicine, 2019) is neither a meta-analysis. Please change them to the proper citations.

4.)    Page 4, line 140 and Table 1 (Lee et al, 2012, Ref Nr 23) you state that included subjects in this article were healthy, however according to the original publication ‘Inclusion criteria were hypercholesterolaemic patients aged 18–60 years, who had been receiving statins as cholesterol-lowering therapy for at least 3 months before the commencement of the study.’ Please correct this in the whole article. I also think that you have mixed the numbers of two other publications: Oliveira-de-Lira (Nr 12) included women with abdominal adiposity (but NOT hyperlipidaemia), while Jones et al (Nr 11) included subjects with obesity AND high plasma TG.

5.)    Since I have found several discrepancies (some of them mentioned above in point 2-4) between the data you describe and the actual data published in the original articles, I ask you that all the data provided in Table 1 and the Study characteristics (lines 137-155) you describe in the text should be reviewed point by point with the original publications.

6.)    This SR was performed in 2022, so I don’t understand why you cite this old version of Cochrane Handbook (v5.1.0, updated in March 2011). There are four more up-to-date electronic versions (v5.2 from 06.2017; v6.0 from 07.2019, v6.1 from 09.2020; v6.2 from 02.2021) and the most recent one: v6.3 (updated in 02.2022). Even the printed edition is from 2019. As the literature search was performed in 12.2022, the most recent version should have been applied. Please clarify this!

7.)    There is a Cochrane SR in the topic omega-6 fats in the prevention of cardiovascular diseases, and they also publish data about blood lipids. I think this article should be an essential part of the discussion (or introduction) section: Hooper L, Al-Khudairy L, Abdelhamid AS, Rees K, Brainard JS, Brown TJ, Ajabnoor SM, O'Brien AT, Winstanley LE, Donaldson DH, Song F, Deane KH. Omega-6 fats for the primary and secondary prevention of cardiovascular disease. Cochrane Database Syst Rev. 2018 Nov 29;11(11):CD011094. doi: 10.1002/14651858.CD011094.pub4. PMID: 30488422; PMCID: PMC6516799.

8.)    Discussion: lines 362-239 doesn’t seem to be part of this section. Please delete it.

9.)    Page 1, line 43: the fatty acid is called alpha-linolenic acid, so either insert the Greek letter ‘α’ or write the word ‘alpha’.

10.) Doi is missing for references Nr 15 (Pang et al), Nr 20 (Iggman et al), Nr 23 (Lee et al), Nr 28 (Junker et al), Nr29 (Abbey et al), Nr 58 (Mansink et al), Nr 63 (Pedersen et al), Nr 68 (Plat et al) and for Ref. 21 doi is doubled (Karupaiah et al). Please correct them. Also please check the whole manuscript, because there are unnecessary words in some sentences which made these sentences meaningless.

The quality of English language is quite fine, however, there are some parts of the text where style or unnecessary words should be revised.

Reviewer 2 Report

The manuscript foods-2360780 describes a meta-analysis approach to evaluate the effect of dietary linoleic acid on blood lipid profiles including triglycerides, total cholesterol, high-density lipoprotein cholesterol, and low-density lipoprotein cholesterol. In general, the subject of the current manuscript is surely worthy of investigation. However, the manuscript needs extensive revisions, and several concerns need to be addressed as follows:

-          Throughout the manuscript, the writing style should be formal from the third-person perspective. Do not use “we” (e.g. lines 59, 66, 120… etc ) or “our” (e.g. lines 23, 138, 207… etc ). 

-          L23-24: The conclusion section should be rewritten, as in the current form, it is only a general statement. The conclusion should answer the aim of the study.

-          L47-50: “ ……revealed potential long-term advantages of LA intake in lowering the risk of death, as well as reducing blood glucose and lipids. Supplemental LA exerted a beneficial effect on blood lipid profiles in participants with obesity” Specify the mode of action.

-          L132-133: Please add the criteria used for articles selection here.

-          The use of the " plasma " term is not accurate as some studies used the serum, such as Rallidis et al [34]. So, it is highly recommended to replace " plasma" with "blood" throughout the manuscript.

-          Table 1: replace “Study and year” with “Reference”. Also, “Spanish” and “Malaysian” should be “Spain” and “Malaysia”

-          Table 2: p-values are missed. In the footnote, it is “No.” or “N”?

-          L236-238: It appears as a reviewer comment. please revise.

-          Discussion section: The author should give more discussions on the mode of action of dietary linoleic acid to regulate blood lipid profile and present the lessons learned from the state of the science and challenges in this field to show the manuscript's contribution more clearly in the practical field.

Moderate editing of the English language is required.

Round 2

Reviewer 1 Report

The whole text has been revised, and the MS improved a lot. In its current form, the manuscript is more logically structured and easier to follow. I also accept the answers to my questions. I have a major comment: throughout the whole MS you mix up the terms fats and fatty acids and use them as synonyms, but they mean different things. Fatty acids are components of different lipid classes, like triacylglycerols, sterol esters, sphingomyelins. The term fat is mainly used for triacylglycerols (almost exclusively animal-derived storage lipids), that mainly consist of saturated fatty acids and therefore they are solid on room temperature.

I also have some minor comments:

1.)    Line 58 “that” is unnecessary, please delete it or reword the sentence to make sense.

2.)    In many places a space is missing between bracket and text (e.g. line 45, 61). Please correct them in the whole MS.

3.)    Lines 67-68: either different study design/intervention or differences in study design/intervention. Please correct both.

4.)    Lines 68-70, please reword the sentence to make sense.

5.)    Line 73: increasing n-6 fats: is this dietary intake of n-6 fatty acids? Please clarify it.

6.)    Lines 79-80: trans fatty acid (TFA). FFA is the abbreviation for free fatty acids.

7.)    There is a discrepancy between text (line 174) and figure 1: total of 3670 vs. 3700 publications. Please correct it.

8.)    Line 198: unit for BMI is missing from the text. Please include it after the numbers.

9.)    Hyphen is unnecessary in line 199 (se-lected), 205 (supplemen-tation) and 206 (sun-flower). Please delete them.

10.) Line 251: if you included 40 eligible studies, how can 42 of them report data about TC concentrations?

11.) Line 255: space is missing before n-3.

12.) Line 324: please use the abbreviation CVD.

13.) Line 325: fatty acids are not food, but food components or nutrients. Please correct it.

14.) Lines 331-332: Please use only one form for fatty acid group: change omega-6 to n-6 (and later also: change all omega-6 to n-6).

15.) Line 333 LA is and essential fatty acid. Please don’t mix fats with fatty acids!

16.) Lines 335-336: do you mean by FFA free fatty acids? Abbreviation is not yet introduced.

17.) Line 340: ‘e’ is missing from evaluate

18.) Line 348: ‘effectively effect’ what do you mean with this?

19.) Line 393 units are missing (for age and BMI)

20.) Lines 398-399 either patients with normolipidemia or normolipidemic patients. Please correct it.

21.) Lines 406-407: verb is missing from the sentence. 

In the MS there are several grammatical and word-use problems, in some sentences verb is missing or the sentence is meaningless. Please check the English style of the whole MS.

Reviewer 2 Report

The authors have satisfactorily revised the manuscript.

Author Response

Thank you very much for your comments on our manuscript. Those comments are all valuable and very helpful for revising and improving our paper, as well as the important guiding significance to our researches.